# Could an Internet-Based Foot–Ankle Therapeutic Exercise Program Modify Clinical Outcomes and Gait Biomechanics in People with Diabetic Neuropathy? A Clinical Proof-of-Concept Study

**DOI:** 10.3390/s22249582

**Published:** 2022-12-07

**Authors:** Ronaldo H. Cruvinel-Júnior, Jane S. S. P. Ferreira, Jady L. Veríssimo, Renan L. Monteiro, Eneida Y. Suda, Érica Q. Silva, Isabel C. N. Sacco

**Affiliations:** 1Department of Physical Therapy, Speech, and Occupational Therapy, School of Medicine, University of São Paulo, Rua Cipotânea, 51—Butantã, São Paulo 05360-160, SP, Brazil; 2. Department of Biological and Health Science, Federal University of Amapá, Rod. Juscelino Kubitschek, km 02—Jardim Marco Zero, Macapá 68903-419, AP, Brazil; 3Postgraduate Program in Physical Therapy, Ibirapuera University, Av. Interlagos, 1329—Chácara Flora, São Paulo 04661-100, SP, Brazil

**Keywords:** diabetic neuropathies, exercise therapy, foot-related exercises, eHealth, rehabilitation technology, proof of concept

## Abstract

Previous studies have shown the efficacy of foot–ankle exercises in people with diabetic peripheral neuropathy (DPN), but the quality of evidence is still low. This proof-of-concept study pursues preliminary evidence for potential clinical and gait biomechanical benefits from an internet-based foot–ankle therapeutic exercise program for people with DPN. We randomized 30 individuals with DPN (IWGDF risk category 1 or 2) into either the control group (CG) receiving the usual care or the intervention group (IG) receiving the usual care plus an internet-based foot–ankle exercise program, fully guided by the Sistema de Orientação ao Pé Diabético (SOPeD; translation: Diabetic Foot Guidance System) three times per week for 12 weeks. We assessed face-to-face clinical and biomechanical outcomes at baseline, 12 weeks, and 24 weeks (follow up). Participants had good adherence to the proposed intervention and it led to only mild adverse events. The IG showed improvements in the ankle and first metatarsophalangeal joint motion after 12 and 24 weeks, changed forefoot load absorption during foot rollover during gait after 24 weeks, reduced foot pain after 12 weeks, and improved foot function after 24 weeks. A 12-week internet-based foot–ankle exercise program using the SOPeD software (version 1.0) has the potential to reduce foot pain, improve foot function, and modify some important foot–ankle kinematic outcomes in people with DPN.

## 1. Introduction

It is estimated that between 12 and 50% of people with diabetes have some degree of diabetic peripheral neuropathy (DPN), which is the most prevalent chronic complication 10 to 15 years after a diabetes diagnosis [1,2]. DPN compromises the structure and functioning of the peripheral nerves, which, in turn, causes sensorimotor disorders [3]. As a result of these disorders, musculoskeletal and biomechanical alterations arise in daily living activities, affecting the quality of motion and functional performance. The most common musculoskeletal alterations are changes in the mechanical properties of joint tissues due to the accumulation of advanced glycation products [4], increased stiffness and reduced range of motion (ROM) in distal joints [5,6], loss of lower-limb muscles strength [7,8], and atrophy of the foot–ankle intrinsic and extrinsic muscles [9,10,11,12]. Biomechanical alterations arise from progressive musculoskeletal changes that compromise proper foot rollover during gait, increasing the loads on the plantar surface [13,14,15], which, in turn, increases the risk for foot ulcers [13,14,15,16,17,18].

From the 2000s to the present day, there have been significant advances in the investigation of the effects of therapeutic exercises targeting the main musculoskeletal and biomechanical deficits of the foot–ankle in people with DPN. These studies have proven the efficacy of these exercises for reducing DPN symptoms and increasing foot–ankle ROM, although the quality of evidence is still low [19]. Therefore, there is still room for investigation of the efficacy of different exercise approaches on biomechanical and clinical outcomes as well as the evaluation of new outcomes, such as foot kinematics and foot–ankle kinetics, which have not yet been addressed [20,21,22]. Although the most recent guidelines from the International Working Group on the Diabetic Foot (IWGDF) [23] incorporated foot–ankle exercises as a new intervention to treat modifiable risk factors for ulceration, this kind of exercise is still poorly recognized among clinicians and not yet widely adopted among rehabilitation professionals. Thus, only a few patients have had the opportunity to benefit from this recommended intervention.

Recently incorporated in healthcare contexts due to the COVID-19 pandemic, telerehabilitation and web-based therapeutic interventions have emerged as promising strategies for including foot–ankle exercises in people’s daily routine because of their convenience, reduced costs, and accessibility [24,25,26,27]. This e-health intervention could help address the adherence and compliance problems usually reported among these patients, such as low self-motivation, treatment costs, and longer treatment duration [28]. Inspired by this scenario, our research group developed and validated the Sistema de Orientação ao Pé Diabético (SOPeD; translation: Diabetic Foot Guidance System; www.soped.com.br, accessed on 19 May 2022), which customizes foot–ankle exercises and stimulates self-care and self-management actions in people with diabetes and DPN. This free software has emerged as an alternative to face-to-face physiotherapy to treat musculoskeletal disorders arising from diabetes and DPN [29]. The proposed clinical proof-of-concept study pursues preliminary evidence for the potential efficacy of a 12-week internet-based foot–ankle therapeutic exercise program for people with diabetes and DPN to promote clinical and gait biomechanical changes.

## 2. Method

### 2.1. Study Design

This clinical proof-of-concept study is part of a full randomized controlled clinical trial, the Foot Care I (FOCA-I). Reporting is based on the Consolidated Standards of Reporting Trials extension for web-based and mobile-health interventions (CONSORT-EHEALTH) [30]. The main trial was approved by the ethics committee of the School of Medicine of the University of Sao Paulo (CAAE: 90331718.4.0000.0065) and was prospectively registered at ClinicalTrials.gov on 8 July 2019 (NCT04011267). The full protocol is detailed elsewhere [31]. This clinical proof-of-concept study and the main trial were designed as a parallel-group, two-arm, superiority trial with a 1:1 allocation ratio.

Allocation to the control group (CG; n = 15) and intervention group (IG; n = 15) was performed after acquiring baseline data using a randomization sequence [32] that was kept in opaque and sealed envelopes after having been organized into blocks by an independent researcher. Only the physiotherapist responsible for the intervention’s prescription was aware of group allocation. The Brazilian General Law for the Protection of Personal Data (No. 13.709/2018) was respected by encoding the names of the participants and keeping the personal data confidential before, during, and after the study. The study statistician and the two other researchers responsible for all clinical and biomechanical assessments were blinded to the allocation. The participants were assessed at baseline, after 12 weeks of intervention, and after 24 weeks from baseline (follow-up measure) at the Physical Therapy Department of the School of Medicine of the University of São Paulo. 

Data for this clinical proof-of-concept study were collected between September 2019 and September 2021 (Figure 1). Participants were recruited from the patient database of the Endocrinology Outpatient Clinic of the Hospital das Clínicas, School of Medicine, University of São Paulo.

### 2.2. Participants

The first 30 adults from the FOCA-I trial were included in this study. Adults of both sexes, between 18 and 65 years old, and with a clinical diagnosis of type 1 or 2 diabetes and DPN (IWGDF risk category 1 or 2) were considered. The first contact with the patient was made by telephone, and the potential participants were assessed at the biomechanics laboratory to confirm the eligibility criteria: independent walking ability, access to the internet and ability to use electronic devices (e.g., computer, mobile phone, or tablet), and DPN severity score above 2 confirmed by the Decision Support System for Classification of Diabetic Polyneuropathy [33] (www.usp.br/labimph/fuzzy, accessed on 19 May 2022). This system is based on fuzzy logic and three input variables: signs and symptoms extracted from the Brazilian version of the Michigan Neuropathy Screening Instrument (MNSI-BR), vibration sensitivity evaluated by a tuning fork (128 Hz), and tactile sensitivity measured by a 10 g monofilament.

Participants with any of the following criteria were not included: amputation of any foot parts; an ulcer that had not healed for at least 6 months and/or an active ulcer; history of surgical procedures in the foot, ankle, knee, or hip or indication of surgery or arthroplasty; arthroplasty and/or orthosis of lower limbs or indication of lower-limb arthroplasty throughout the intervention period; diagnosis of other neurological disease outside diabetes etiology; dementia or inability to provide consistent information; receiving any physiotherapy or offloading devices throughout the intervention; use of assistive devices for walking; and major vascular complications and/or severe retinopathy as determined from medical files. The principal investigator explained to each eligible participant all stages of the study, possible risks, and expected benefits. Upon agreeing to participate, they were asked to sign an informed consent form.

### 2.3. Treatment Arms

CG participants received the usual care, including treatment recommended by the medical team, standard pharmacological treatment, and self-care guidelines based on the IWGDF [34]. According to IWGDF recommendations, the use of therapeutic footwear or a custom-made insole is mandatory for patients with high ulcer risk (IWGDF category 3). As there is no mandatory prescription for low and moderate ulcer risk patients (IWGDF categories 1 or 2), the use of therapeutic footwear or a custom-made insole was not prescribed for the participants included in this study. These self-care guidelines, adjusted for our study, were printed on a flyer given to all participants and included educational orientations (Appendix A—flyer of self-care based on IWGDF).

IG participants followed the usual care plus an internet-based foot–ankle exercise program guided by the SOPeD. The exercise program had a total of 36 sessions (three sessions per week for 12 consecutive weeks) and each session, including eight exercises, lasted 20 to 30 min and was performed at a time convenient for the participant. The comprehensive therapeutic exercise protocol is detailed elsewhere [29,31], but in summary, the SOPeD includes a total of 104 functional, stretching, and strengthening exercises of the extrinsic and intrinsic foot muscles (Figure 2A). The progression for each exercise in intensity and complexity was customized based on the individual’s perceived effort as determined by an algorithm (Figure 2B). If the effort category selected by the participant was “not tiring” or “a little tiring” the software increased the exercise intensity at the next session. If the user selected “tiring” the software advanced to the next intensity level after two sessions at the current exercise intensity. If “very tiring” was selected, the software decreased the difficulty/intensity and returned to the previous level. No changes were made to the software content, and the intervention protocol algorithm remained the same throughout the clinical trial. The SOPeD includes gamification components [35] to increase adherence and encourage users to continue exercising (Figure 2C).

The first session was delivered face to face by the physiotherapist to explain the use of the software, ensure the correct execution of the exercises, and deliver a kit with materials for performing the exercises (cotton balls, a towel, a pencil, mini elastic bands, balloons, light- and moderate-resistance elastic bands, a massage ball, and finger separators) to the IG participants. The main physiotherapist supervised all of the other 35 sessions remotely via the SOPeD interface. Participants in the intervention group received access to use SOPeD that aims to provide self-care and allows the user to choose the best and most convenient time to carry out the exercise sessions. The exercise sessions were not monitored synchronously, but the main researcher could have access to SOPeD administrator, at any time, to monitor how often they accessed the software and how many exercise sessions were performed by each participant. Participants were instructed to stop exercising and communicate with the main researcher if they experienced cramps, moderate to severe pain, excessive fatigue, or any other condition that caused discomfort. 

Every two weeks, a physiotherapist called the participants to check on their performance, difficulties, and the occurrence of any adverse events. If the IG participant did not access the software for more than three sessions in a row, an email was automatically sent, and the main researcher made a phone call to those participants who did not respond to email reminders. During the follow-up period, IG participants were encouraged to follow the same exercise schedule set by the SOPeD program until the end of the study (24 weeks), but they were not remotely monitored.

The average number of sessions completed (36 sessions total) by IG participants was used to calculate the adherence to the program [36]. The number of all completed sessions was obtained from the SOPeD user databank and computed, even if the participant did not complete the full set of exercises in a given session.

### 2.4. Outcome Measures

The primary outcomes were DPN symptoms and severity, as both were subject to improvements after therapeutic foot–ankle exercises [21,37], are important modifiable factors, and DPN severity is an outcome assessed using a more comprehensive score than just using symptoms as outcomes, which includes DPN symptoms and signs (vibration and tactile sensitivities). Symptoms were measured using the MNSI-BR, which comprises 15 questions with a total score ranging from 0 to 13 (with 13 representing the worst DPN) [38]. DPN severity was determined by the Decision Support System for Classification of Diabetic Polyneuropathy [33], the scores of which range from 0 to 10, with a higher fuzzy score indicating more severe DPN. The secondary outcomes included foot health and functionality, toe and hallux strength, plantar pressure distribution, and joint kinetics and kinematics during gait. 

Foot health and functionality were assessed using the Brazilian version of the Foot Health Status Questionnaire, for which the scores range from 0 to 100 points, where 100 represents the best condition and 0 the worst [39]. Hallux and toe isometric strength was assessed standing using a pressure platform (emed-q100; novel GmbH, Munich, Germany) according to the protocol by Mickle et al. [40]. Maximum force (N) was normalized by body weight and analyzed for the hallux and toe areas separately using a standard mask from novel Multimask software v.9.35 (novel GmbH). The average of the four trials (right and left side) was used for statistical purposes following the rationale described by Menz [41].

Peak pressure, pressure–time integral, and contact area during gait were acquired by the emed-q pressure platform at 100 Hz. The participants walked six times barefoot over the platform at a self-selected comfortable speed. Seven plantar regions of interest (heel, midfoot, medial forefoot, central forefoot, lateral forefoot, hallux, and toes) were assessed by a geometric mask using the novel software. The average of the six trials (right and left side) was used for statistical purposes [40].

The foot–ankle kinematic parameters were recorded using eight infrared cameras at 100 Hz (Vicon VERO; Oxford Metrics, Oxford, UK). Forty-two passive reflective markers (9.5 mm in diameter) were positioned on both lower limbs following the Plug-In Gait and Oxford Foot Model [42] setup protocols. Ground reaction forces for the joint moment calculations were acquired by a force plate (AMTI OR-6-1000; AMTI, Watertown, MA, USA) with a sampling frequency of 1 kHz. A 16-bit analog-to-digital converter was used to synchronize and sample the kinematic and ground reaction force data.

Participants were instructed to walk at a comfortable self-selected speed along a 10 m track with a maximum variation of 5% between measurements. The speed was monitored by two photoelectric cells (Model Speed Test Fit; CEFISE, Nova Odessa, Brazil) to ensure that the same speed was maintained in all assessments (baseline, 12 weeks, and 24 weeks). Five valid steps were acquired during gait and the average (right and left side) was used for statistical purposes [40].

The Motion Capture Nexus 2.6 software (Oxford Metrics) was used for automatic digitizing, three-dimensional reconstruction of marker positions, kinematic and kinetic data filtering, and joint moment calculations. Kinematic data were processed using a zero-lag second-order low-pass filter with cutoff frequency of 6 Hz. Ground reaction force data during walking were processed using a zero-lag low-pass Butterworth fourth-order filter with cutoff frequency of 50 Hz. The bottom-up inverse dynamics method was used to calculate the ankle joint moment in the sagittal plane. For the calculation of the ankle power, the calculated joint moment and angular velocity of the ankle in the sagittal plane were considered. All discrete variables from the angles and moments time series were calculated with the open-source Python package pyCGM2 (http://www.pycgm2.github.io, accessed on 19 May 2022), which replicates the Vicon Plug-In Gait protocol and the Oxford Foot Model Plug-In.

### 2.5. Statistical Analysis

The main trial sample size was calculated using two important outcomes for patients with DPN. Considering the primary outcome (DPN symptoms), a medium effect size (0.52) was adopted and, for the secondary outcome (peak pressure at forefoot), a small effect size (0.20) was adopted. In order to obtain the largest sample size, the smallest effect size (0.20) was used. A statistical design of F-test repeated measures and interaction between and within factors with two repeated measures and two study groups, a statistical power of 0.80, an alpha of 0.05, and an effect size of 0.20 were used for the sample size calculation. The resulting sample size was 52 individuals. A final sample size of 62 patients was then chosen after estimating a drop-out rate of 20%. The current study presents the findings for the first 30 participants.

According to normal data distribution (Shapiro–Wilk test, *p* > 0.05), the baseline participants’ characteristics were reported as means and standard deviations, numbers and percentages, or medians and interquartile ranges (IQRs). An intention-to-treat approach and the Generalized Estimating Equation (GEE) method were used with an exchangeable correlation structure and the following fixed factors: groups (CG and IG), assessment timepoint (baseline, 12 weeks, and 24 weeks), and the interaction effect (group–time). The Gamma distribution was used to select the GEE model based on the quasi-likelihood under the independence model criterion, resulting in a better model fit. Between-group differences at 12 and 24 weeks and their 95% confidence intervals were reported [43]. All statistical analyses were carried out using SPSS v.22.0 (IBM, Armonk, New York, NY, USA) with a significance level of 5%.

## 3. Results

Participant flow, attendance at follow-up assessment visits, and reasons for dropout are presented in Figure 1. At baseline, the groups were similar for all characteristics and outcomes assessed (Table 1). In the IG, 14 participants (93.3%) completed the 12-week internet-based foot–ankle therapeutic exercise program and the adherence was 62.0%. The dropout rate in the IG, that is, the number of participants who did not attend both assessments at 12 and 24 weeks, was 6.6% (one participant). Unfortunately, some participants in both groups did not attend the 12- and 24-week follow-up visits due to the COVID-19 pandemic (lost to follow up). The lost-to-follow-up rate at 12 weeks was 10% for the whole sample, 13.3% in the CG (two participants), and 6.6% in the IG (one participant). The lost-to-follow-up rate at 24 weeks was 20% in the CG (three participants) and 40% in the IG (six participants). In addition, two patients from the IG reported mild adverse effects of the intervention, which were delayed onset muscle soreness and cramping in the foot muscles. None of the participants withdrew from the trial due to adverse effects.

After 12 and 24 weeks, the IG displayed no significant interaction effects for any clinical or plantar pressure outcomes but did show group and time effects (Table 2 and Table 3). The between-group analysis showed a significant reduction in foot pain in the IG compared to the CG at 12 weeks (group effect: *p* = 0.023, post hoc: *p* = 0.004) and within the IG after 12 weeks (post hoc: *p* = 0.001) and 24 weeks (post hoc: *p* = 0.010) compared to baseline (time effect: *p* = 0.002). An improvement in the foot function was also observed in the IG compared to the CG (group effect: *p* = 0.083, post hoc: *p* = 0.040) and within the IG after 12 weeks (post hoc: *p* = 0.006) and 24 weeks (post hoc: *p* = 0.012) compared to baseline (time effect: *p* = 0.001) (Table 2). The pressure-time integral in the medial forefoot was significantly increased in the IG compared to the CG at 24 weeks (group effect: *p* = 0.004, post hoc: *p* = 0.048) (Table 3).

Interaction effects were identified after 12 and 24 weeks for gait kinetics and kinematics (Table 4). The ankle plantar flexion angle at push-off was significantly increased in the IG compared to the CG (interaction effect: *p* = 0.049) at 12 weeks (post hoc: *p* = 0.013) and 24 weeks (post-hoc: *p* = 0.014). The within-group analysis showed a significant increase in the IG participants after 12 weeks (post hoc: *p* = 0.001) and 24 weeks (post hoc: *p* = 0.001) compared to baseline (time effect: *p* = 0.001). The within-group analysis showed changes in the hindfoot to tibia peak angle in the IG after 24 weeks compared to baseline (time effect: *p* = 0.017, post hoc: *p* = 0.033) and this improvement was greater in the IG compared to the CG after 24 weeks (interaction effect: *p* = 0.038, post hoc: *p* = 0.009). The hallux to forefoot ROM increased in the IG compared to the CG at 24 weeks (group effect: p =0.028, post hoc: p=0.003), and the hallux to forefoot peak angle increased in the IG compared to the CG (group effect: *p* = 0.049) at 12 weeks (post hoc: *p* = 0.016) and 24 weeks (post hoc: *p* = 0.021). Finally, at 12 weeks, the maximum arch height (group effect: *p* = 0.049, post hoc: *p* = 0.015) and minimum arch height (group effect: *p* = 0.044, post hoc: *p* = 0.020) were smaller in the IG compared to the CG.

## 4. Discussion

This clinical proof-of-concept study aimed at gathering preliminary evidence for the potential clinical and gait biomechanical benefits of the SOPeD, an internet-based foot–ankle therapeutic exercise program for people with diabetes and DPN. Some beneficial effects in terms of foot function, pain, and foot–ankle kinematics were revealed. After 12 weeks of intervention, participants had less foot pain intensity and frequency, improved foot function, increased ankle and first metatarsophalangeal (MTP) joint motions, favorably altered foot arch motion, and increased forefoot loads during gait.

The pain reduction and function improvement could be explained by the improved foot–ankle ROM, as better joint mobility positively affects the overall foot functionality and, thus, pain. Because there was a reduction in foot pain after 12 weeks and an improvement in foot function only after a longer period of 24 weeks, we believe that the improvement in foot function is not necessarily related to pain reduction but is the result of changes in other functional components due to the exercises, such as improved foot–ankle joint mobility. A feasibility study from this trial showed a significant improvement in foot pain in the IG [27], and this proof-of-concept study confirmed these findings by demonstrating that an internet-based foot–ankle intervention achieved this secondary outcome.

Some studies have shown that individuals with diabetes and DPN have a reduction in the ankle and first MTP joint mobility, and these movement restrictions may contribute to ulcerations at all forefoot locations [6,44,45,46,47]. Due to their reduced ankle mobility, people with DPN appear to pull their legs forward during the push-off phase, mainly using the hip flexor muscles, which is known as the hip strategy. In contrast, the ankle strategy, which is characterized by propelling the body forward while relying on the plantar flexor muscles, is seen in the gait of people without diabetes and DPN [48,49]. We found that this 12-week internet-based foot–ankle exercise program was effective in improving some of the foot–ankle joint motion (hindfoot to tibia angle, plantarflexion at push-off, and hallux to forefoot ROM and angle), which may help the subjects to prioritize the use of the physiological ankle strategy instead of the hip strategy and, thus, promote a more physiological foot rollover.

Recent studies have shown positive results for the improvement in ankle ROM and first MTP joint mobility using foot-related exercise programs with group-based [37,50] and home-based [20,21,51] approaches. Even though those interventions were either conducted face to face with the physiotherapist or at home with only the guidance of videos or booklets, this still corroborates our results. None of these previous interventions used a rehabilitation technology to promote foot–ankle exercises targeting the main musculoskeletal deficits in the lower limbs. Thus, performing foot–ankle exercises with the support of a web-based software—the SOPeD—has the potential to be as effective as performing face-to-face (group-based) or home-based therapeutic programs.

The intervention promoted an increase in the pressure-time integral at the medial forefoot in the IG after 24 weeks, which can be attributed to the gains obtained in the foot–ankle and first MTP mobilities. Greater mobility of the medial forefoot region, including the first MTP joint, is desirable during the mid-stance phase of gait to better adjust the foot to the ground in the pronation movement expected in this phase, thus, favoring the propulsion through the first ray of the foot. As a consequence, changes in the plantar pressure distribution in this foot area would reveal greater anterior support during the mid-push-off phase of gait. Furthermore, factors affecting foot biomechanics, such as reduced joint range of motion and foot deformity, have been linked to changes in plantar pressure distribution [52], which is consistent with our findings of an improvement in hallux to forefoot ROM and peak angle. Additionally, the increase in the pressure-time integral at the medial forefoot could have potentially contributed to a more physiological foot rollover. Our findings agree with the results of Sartor et al. [53], who also found an increase in the pressure-time integral at the medial and lateral forefoot and hallux and attributed this change to an improved foot rollover into a more physiological process and a better functional condition of the foot–ankle complex. While attention is usually given to peak pressure and pressure-time integral reductions as targets to reduce the risk of ulceration, these variables only represent vertical loading during a very short time in the stance phase and are not optimal variables for describing changes in the whole foot rollover process, which should be the main aim of rehabilitation strategies, such as foot-related exercise programs. Although the increased pressure-time integral at the medial forefoot in the IG participants might represent a functional improvement in gait, attention must be given to keeping plantar loadings under a safe pressure range in foot areas at higher risk for ulceration [54].

The proposed 12-week exercise program has the potential to change the maximum and minimum arch height during gait, with the IG showing lower values after the intervention. The plantar arch should be flexible in response to gait loads, allowing foot-joint adjustments to dampen impacts via multiple mechanisms, such as stiffness and power absorption, but it should also be rigid enough to allow for propulsion during the push-off phase [55]. Our exercise program may have improved the plantar intrinsic muscles’ ability to provide force-dependent changes in the MLA and facilitate efficient foot-to-ground contact during walking [56,57].

An analysis of 101 papers conducted as part of a systematic review with the goal of analyzing adherence to web-based interventions revealed an average adherence rate of 50%, confirming that non-adherence is a problem with web-based interventions. The level of adherence varied greatly, with six programs scoring less than 10% and only five interventions achieving 90% or more [58]. The gamification principles, a wide variety of exercises to avoid monotony, and the short duration of the exercise sessions were the strengths of the foot–ankle therapeutic exercise program used in this study, which may have contributed to maximizing the adherence (62%) and minimizing the compliance problems commonly reported in this population [28]. In this study, the reported reasons for not following the online program were internet problems and a broken cell phone.

One of the clinical goals of this proof-of-concept study was to improve participants’ self-management, which can be affected by the individual’s education level [59]. Therefore, having only 6.6% of the participants with a lower level of education may have contributed to our positive results. The study was carried out with participants with ulcer risk (IWGDF categories 1 or 2) and presented positive results that we believe may even contribute to the prevention of the development of foot ulcers and amputations. However, it should be noted that further studies are needed with patients with ulcer risk (IWGDF category 3), since we did not test the intervention in this population. This study also revealed the potential therapeutic effectiveness of our program, which was similar to other previously studied foot-related interventions using face-to-face strategies and emphasizes the importance of an internet-based exercise program as a low-cost, convenient, and easily accessible tele-rehabilitation strategy for people with DPN. These results suggest that this intervention has benefits for people with DPN (low and moderate ulcer risk) and, therefore, its use in clinical practice is promising.

Although the proposed intervention was superior to the usual care and demonstrated the potential to modify some biomechanical and clinical outcomes, such as foot pain, it did not improve the primary outcomes (DPN severity and symptoms, such as burning pain, muscle cramps, and prickling feelings). Our study did not monitor participants’ blood glucose and glycated hemoglobin levels, which may have influenced the primary outcome results. An increase in blood glucose levels may contribute to a greater manifestation of DPN symptoms, which, in turn, would have also influenced the DPN severity, as the fuzzy classification took into account MNSI-BR scores. In addition to the above-mentioned limitation, because no formal calculation of statistical power is performed in proof-of-concept studies, this preliminary analysis should be interpreted with caution.

## 5. Conclusions

This study found that a 12-week internet-based foot–ankle exercise program using the SOPeD software had moderate adherence among participants and has the potential to reduce foot pain, improve foot function, increase foot–ankle and first MTP joint motion, and change forefoot load absorption during foot rollover during gait in people with DPN.

## Figures and Tables

**Figure 1 sensors-22-09582-f001:**
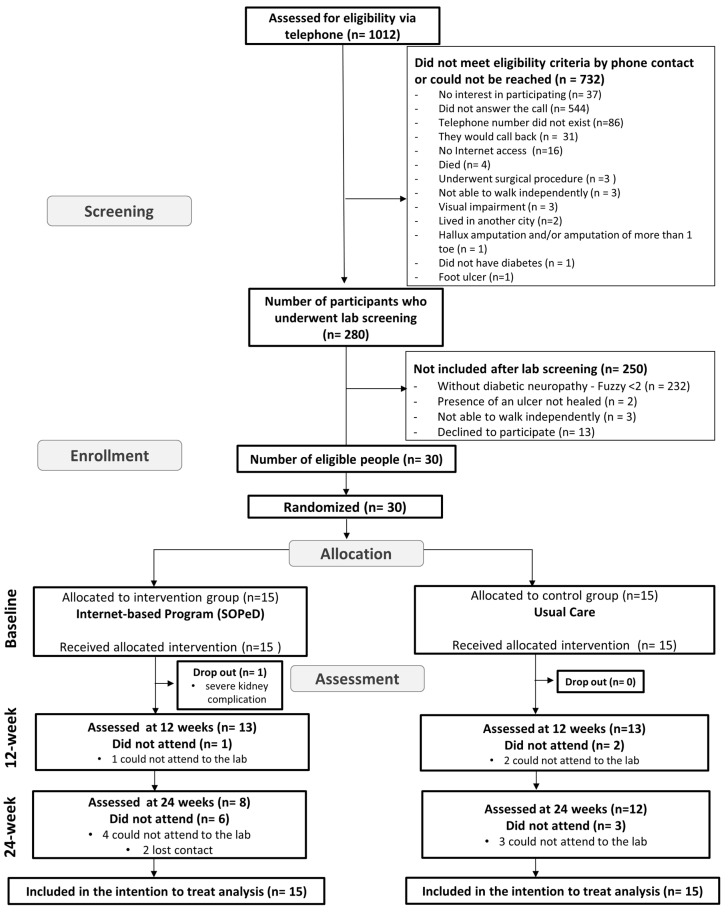
Flowchart of recruitment, assessment, and follow-up process of the proof-of-concept study.

**Figure 2 sensors-22-09582-f002:**
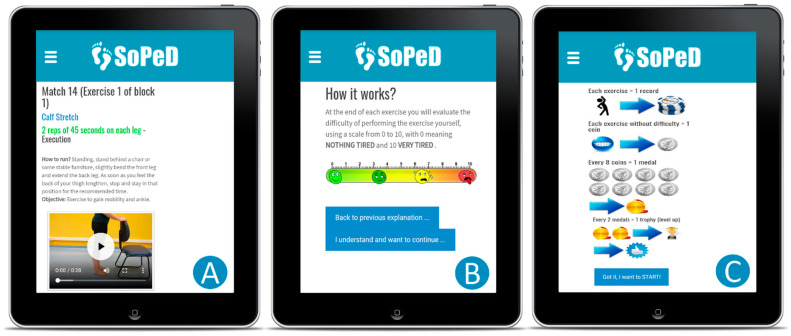
Sistema de Orientação ao Pé Diabético (SOPeD; translation: Diabetic Foot Guidance System). (**A**) Layout of the exercises page with video, audio, and written instructions. (**B**) Perceived effort scale to be completed after each exercise performed. (**C**) Exercise protocol rules with gamification components.

**Table 1 sensors-22-09582-t001:** Clinical, demographic, and anthropometric outcomes at baseline for the control and intervention groups.

	Control Group (n = 15) Mean (SD)	Intervention Group (n = 15) Mean (SD)
Age (years)	56.5 (9.9)	51.1 (10.2)
Body mass (kg)	81.5 (18.6)	80.0 (16.5)
Height (cm)	161.0 (0.1)	169.0 (0.1)
Body mass index (kg/m^2^)	31.8 (8.1)	28.0 (5.1)
Sex (Female) (n, %)	(F = 10/66.6%)	(F = 8/53.3%)
Type 2 Diabetes (number of participants, %)	14 (93%)	13 (86.6%)
Time of onset of diabetes (years)	10.8 (7.4)	18.8 (11.8)
Education (number of participants, %)		
Elementary education incomplete	0 (0%)	1 (6.6%)
Elementary education complete	2 (13.3%)	0 (0%)
High school incomplete	3 (20.1%)	0 (0%)
High school complete	7 (46.7%)	5 (33.4%)
Higher education incomplete	1 (6.6%)	0 (0%)
Higher education complete	2 (13.3%)	9 (60.0%)
Socioeconomic status (number of participants, %)		
1 to 3 Brazilian minimum salary/month	13 (86.7%)	7 (46.7%)
3 to 5 Brazilian minimum salary/month	2 (13.3%)	1 (6.6%)
Up to 5 Brazilian minimum salary/month	0 (0%)	7 (46.7%)
DPN symptoms (MNSI score)	6.9 (1.5)	7.3 (1.8)
DPN severity (Fuzzy score)	3.5 (1.8)	4.3 (2.3)
Tactile sensitivity (number of areas, Median [IQR])	0 [0–0]	0 [0–1]
Vibration Perception (number of participants, %)		
absent-L	1 (6.6%)	5 (33.3%)
reduced-L	0 (0%)	3 (20%)
absent-R	3 (20%)	3 (20%)
reduced-R	2 (13.3%)	4 (26.6%)
FHSQ (score)		
Foot pain	39.8 (21.3)	50.9 (22.5)
Foot function	56.2 (27.2)	68.7 (24.1)
Shoes	41.7 (37.8)	59.4 (37.8)
Foot health	22.5 (19.8)	15.8 (12.0)
Foot Strength (%BW)		
Hallux	17.0 (6.8)	10.9 (4.1)
Toe	10.3 (5.2)	8.2 (3.8)

Data are presented as mean (SD) or as n or %; and median (interquartile range IQR). Abbreviation: MNSI-Michigan Neuropathy Screening Instrument; L—Left; R—Right; FHSQ—Foot Health Status Questionnaire; DPN diabetic peripheral neuropathy; BW—body weight.

**Table 2 sensors-22-09582-t002:** Estimated mean (standard error; SE), *p*-values from Generalized Estimating Equation (GEE), and between-group mean differences at 12 and 24 weeks (95% confidence interval) of the clinical outcomes for the control and intervention groups.

	Intervention Group (n = 15)			Control Group (n = 15)			Between-Group Difference (CI 95%)		GEE Analysis (*p*-Values)		
Variables	Baseline Estimated Mean (SE)	12-Week Estimated Mean (SE)	24-Week Estimated Mean (SE)	Baseline Estimated Mean (SE)	12-Week Estimated Mean (SE)	24-Week Estimated Mean (SE)	12-Week (Intervention X Control Group)	24-Week (Intervention X Control Group)	Group	Time	Group X Time (Interaction Effect)
DPN symptoms (MNSI score)	7.3 (0.4)	5.1 (0.6)	5.0 (0.6)	6.9 (0.4)	6.1 (0.6)	5.4 (0.4)	−0.9 (−2.6 to 0.7)	−0.4 (−1.9 to 1.1)	0.545	<0.001	0.152
DPN severity (Fuzzy score)	4.3 (0.6)	3.5 (0.7)	3.7 (0.8)	3.5 (0.5)	3.2 (0.5)	3.3 (0.5)	0.2 (−1.4 to 1.9)	0.3 (−1.5 to 2.2)	0.534	0.096	0.534
FHSQ Foot pain (score)	50.9 (5.6)	72.6 (6.4) *	72.0 (8.7)	39.8 (5.3)	47.7 (5.8) *	51.3 (7.8)	24.9 (8.0 to 41.8) *	20.7 (−43.5 to 2.1)	0.023*	0.002	0.514
FHSQ Foot function (score)	68.7 (6.0)	80.8 (6.6)	84.1 (6.3) ^#^	56.2 (6.8)	69.6 (7.0)	63.3 (7.9) ^#^	11.2 (−7.8 to 30.1)	20.7 (0.9 to 40.5) ^#^	0.083	0.001	0.619
FHSQ Shoes (score)	68.6 (8.4)	61.5 (9.1)	68.3 (10.2)	52.1 (9.7)	60.9 (8.2)	55.1 (7.7)	0.6 (−23.4 to 24.7)	13.2 (−11.8 to 38.2)	0.271	0.976	0.414
FHSQ Foot health (score)	23.7 (1.2)	41.9 (6.3)	43.50 (7.7)	30.7 (4.7)	37.7 (4.5)	42.0 (6.3)	4.2 (−11.1 to 19.5)	1.4 (−18.0 to 21.0)	0.781	0.001	0.234
Hallux strength—(%BW)	10.9 (1.0)	14.4 (1.3)	10.0 (1.1)	14.9 (1.7)	14.7 (1.7)	12.1 (1.1)	−0.3 (−4.4 to 3.8)	−2.0 (−5.0 to 0.9)	0.640	0.002	0.064
Toes strength—(%BW)	8.2 (0.9)	7.9 (1.0)	8.5 (1.1)	10.3 (1.3)	8.7 (1.1)	8.1 (1.4)	−0.8 (−3.7 to 2.1)	0.4 (−3.1 to 4.0)	0.594	0.343	0.207

Abbreviation: MNSI—Michigan Neuropathy Screening Instrument; FHSQ—Foot Health Status Questionnaire; BW—Body Weight. * group effect *p* = 0.023, between-group difference at 12 weeks (post hoc *p* = 0.004). ^#^ group effect *p* = 0.083, between-group difference at 24 weeks (post hoc *p* = 0.040).

**Table 3 sensors-22-09582-t003:** Estimated mean (standard error; SE), p-values from Generalized Estimating Equation (GEE), and between-group mean differences at 12 and 24 weeks (95% confidence interval) of the plantar pressure variables during gait for the control and intervention groups.

		Intervention Group (n = 15)			Control Group (n = 15)			Between-Group Difference (CI 95%)		GEE Analysis (*p*-Values)		
Region of Interest	Variables	Baseline Estimated Mean (SE)	12-Week Estimated Mean (SE)	24-Week Estimated Mean (SE)	Baseline Estimated Mean (SE)	12-Week Estimated Mean (SE)	24-Week Estimated Mean (SE)	12-Week (Intervention X Control Group)	24-Week (Intervention X Control Group)	Group	Time	Group X Time (Interaction Effect)
Toes	Contact Area [cm^2^]	9.96 (0.67)	9.90 (0.65)	8.15 (1.48)	10.86 (0.86)	10.93 (0.90)	10.89 (1.31)	−1.02 (−3.22 to 1.16)	−2.73 (−6.63 to 1.16)	0.173	0.579	0.607
	Peak [kPa]	282.94 (38.86)	350.65 (54.87)	362.50 (75.14)	307.61 (45.61)	302.11 (47.54)	343.86 (49.21)	48.53 (−93.75 to 190.83)	18.63 (−157.42 to 194.70)	0.838	0.194	0.209
	Pressure-time integral [(kPa) * s]	88.80 (14.17)	90.89 (14.19)	97.32 (22.45)	95.52 (15.72)	90.13 (15.12)	117.53 (21.07)	0.76 (−39.88 to 41.41)	−20.21 (−80.57 to 40.14)	0.683	0.385	0.593
Hallux	Contact Area [cm^2^]	7.63 (0.66)	7.94 (0.71)	6.37 (0.40)	8.41 (0.59)	8.34 (0.57)	6.71 (0.49)	−0.39 (−2.19 to 1.39)	−0.33 (−1.59 to 0.56)	0.441	<0.001	0.911
	Peak [kPa]	337.00 (35.28)	378.33 (54.69)	186.89 (26.25)	304.11 (31.19)	361.34 (36.73)	193.88 (20.39)	16.98 (−112.15 to 146.13)	−6.99 (−72.15 to 58.16)	0.765	<0.001	0.731
	Pressure-time integral [(kPa) * s]	100.35 (11.03)	102.16 (20.56)	52.86 (6.89)	89.67 (13.45)	95.86 (11.27)	62.64 (6.75)	6.29 (−39.67 to 52.26)	−9.78 (−28.29 to 9.12)	0.990	<0.001	0.363
Medial forefoot	Contact Area [cm^2^]	18.18 (0.83)	18.39 (0.98)	22.49 (0.84)	19.29 (0.61)	19.53 (0.81)	23.81 (0.92)	−1.14 (−3.65 to 1.36)	−1.32 (−3.78 to 1.14)	0.244	<0.001	0.997
	Peak [kPa]	488.72 (46.36)	502.26 (52.29)	542.34 (43.35)	378.16 (26.13)	421.85 (32.08)	462.98 (28.43)	80.40 (−39.85 to 200.65)	79.36 (−22.25 to 180.97)	0.060	<0.001	0.181
	Pressure-time integral [(kPa) * s]	175.21 (15.40)	169.81 (20.84)	193.26 (13.94) *	134.42 (11.49)	145.85 (12.04)	161.42 (8.09) *	23.96 (−23.22 to 71.14)	31.84 (0.25 to 63.44) *	0.004 *	0.066	0.124
Central forefoot	Contact Area [cm^2^]	21.67 (0.42)	21.05 (0.73)	16.13 (1.20)	22.21 (0.45)	22.35 (0.51)	17.73 (1.31)	−1.30 (−3.05 to 0.44)	−1.59 (−5.09 to 1.89)	0.228	<0.001	0.228
	Peak [kPa]	496.72 (42.08)	473.09 (37.50)	474.92 (41.43)	425.27 (33.56)	496.47 (50.99)	382.98 (28.44)	−23.37 (−147.44 to 100.69)	91.93 (−6.56 to 190.44)	0.278	0.159	0.073
	Pressure-time integral [(kPa) * s]	180.73 (12.57)	159.71 (12.08)	165.24 (12.76)	157.37 (11.41)	169.43 (16.21)	139.13 (9.45)	−9.71 (−49.34 to 29.91)	26.11 (−5.02 to 57.25)	0.341	0.097	0.086
Lateral forefoot	Contact Area [cm^2^]	14.48 (0.73)	12.93 (1.08)	7.82 (0.96)	13.23 (0.96)	14.14 (0.93)	9.39 (0.91)	−1.21 (−4.02 to 1.59)	−1.57 (−4.18 to 1.03)	0.558	0.000	0.128
	Peak [kPa]	286.22 (55.89)	303.63 (64.53)	146.13 (38.34)	216.22 (18.87)	224.48 (26.17)	102.29 (12.62)	79.14 (−57.34 to 215.63)	43.84 (−35.26 to 122.95)	0.155	0.000	0.942
	Pressure-time integral [(kPa) * s]	99.08 (15.76)	97.46 (17.97)	44.78 (13.51)	80.90 (7.27)	81.92 (9.50)	39.52 (5.58)	15.53 (−24.31 to 55.39)	5.25 (−23.40 to 33.92)	0.444	0.000	0.941
Midfoot	Contact Area [cm^2^]	24.54 (0.80)	23.26 (1.25)	29.79 (1.49)	24.58 (1.10)	24.66 (1.08)	32.08 (1.45)	−1.40 (−4.67 to 1.84)	−2.29 (−6.37 to 1.79)	0.428	0.000	0.422
	Peak [kPa]	284.33 (38.70)	284.64 (37.30)	380.68 (43.88)	241.88 (13.83)	242.24 (15.79)	299.93 (23.23)	42.39 (−37.00 to 121.80)	80.75 (−16.57 to 178.07)	0.156	0.000	0.793
	Pressure-time integral [(kPa) * s]	97.75 (16.31)	80.99 (12.51)	124.26 (16.44)	77.57 (6.36)	77.97 (7.71)	106.94 (9.32)	3.01 (−25.79 to 31.82)	17.31 (−19.74 to 54.38)	0.388	0.000	0.083
Heel	Contact Area [cm^2^]	38.19 (1.05)	32.53 (1.17)	28.38 (1.50)	36.72 (1.65)	29.14 (1.95)	27.17 (1.40)	3.38 (−1.09 to 7.86)	1.20 (−2.82 to 5.23)	0.266	0.000	0.506
	Peak [kPa]	238.76 (17.85)	420.95 (33.05)	477.34 (42.44)	217.59 (19.37)	384.48 (29.89)	373.75 (32.45)	36.46 (−50.88 to 123.81)	103.59 (−1.12 to 208.32)	0.097	0.000	0.338
	Pressure-time integral [(kPa) * s]	338.91 (36.91)	130.80 (12.97)	145.55 (14.57)	354.94 (35.86)	126.30 (10.72)	122.06 (10.39)	4.50 (−28.49 to 37.49)	23.49 (−11.59 to 58.57)	0.596	0.000	0.413

* group effect *p* = 0.004, between-group difference at 24 weeks (post hoc *p* = 0.048).

**Table 4 sensors-22-09582-t004:** Estimated mean (standard error; SE), p-values from Generalized Estimating Equation (GEE), and between-group mean difference at 12 and 24 weeks (95% confidence interval) of the foot–ankle kinematics and ankle joint kinetics during gait for the control and intervention groups.

	Intervention Group (n = 15)			Control Group (n = 15)			Between-Group Difference (CI 95%)		GEE Analysis (*p*-Values)		
Variables	Baseline Estimated Mean (SE)	12-Week Estimated Mean (SE)	24-Week Estimated Mean (SE)	Baseline Estimated Mean (SE)	12-Week Estimated Mean (SE)	24-Week Estimated Mean (SE)	12-Week (Intervention X Control Group)	24-Week (Intervention X Control Group)	Group	Time	Group x Time (Interaction Effect)
ANKLE											
Ankle ROM (degree)	23.14 (1.14)	32.53 (1.17)	28.38 (1.50)	22.82 (0.65)	29.14 (1.95)	27.17 (1.40)	3.38 (−1.09 to 7.86)	1.20 (−2.82 to 5.23)	0.630	0.001	0.839
Ankle dorsiflexion at heel strike (degree)	3.14 (0.64)	2.18 (0.40)	3.36 (0.85)	3.13 (0.57)	3.47 (1.09)	4.54 (0.84)	−1.28 (−3.56 to 0.99)	−1.17 (−3.53 to 1.18)	0.249	0.194	0.525
Ankle plantarflexion at push off (degree)	1.73 (0.01)	4.44 (0.78) ^&^	4.14 (0.68) ^&^	2.14 (0.68)	2.20 (0.66) ^&^	2.15 (1.05) ^&^	2.74 (1.19 to 4.28) ^&^	2.40 (1.05 to 3.75) ^&^	0.001	0.001	0.049^&^
Ankle flexor moment at heel strike (Nm/(BM * Height)	−0.04 (0.01)	−0.05 (0.01)	−0.04 (0.02)	−0.03 (0.01)	−0.04 (0.01)	−0.03 (0.00)	−0.01 (−0.05 to 0.02)	−0.01 (−0.04 to 0.01)	0.224	0.419	0.898
Ankle extensor moment at push off (Nm/(BM * Height)	1.36 (0.04)	1.41 (0.03)	1.43 (0.04)	1.30 (0.03)	1.35 (0.04)	1.37 (0.03)	0.06 (−0.03 to 0.16)	0.05 (−0.05 to 0.16)	0.171	0.160	0.948
Ankle peak eccentric power at the push off (W/BM * Height)	2.49 (0.14)	2.58 (0.9)	2.36 (0.15)	2.45 (0.16)	2.48 (0.11)	2.34 (0.15)	−0.1 (−0.56 to 0.31)	−0.2 (−0.46 to 0.25)	0.397	0.198	0.591
OXFORD FOOT MODEL											
Hindfoot to tibia ROM (degree)	23.12 (1.55)	21.68 (1.09)	23.01 (2.76)	23.48 (1.21)	23.15 (1.32)	24.92 (1.25)	−1.46 (−4.83 to 1.90)	−1.90 (−7.86 to 4.04)	0.495	0.410	0.801
Hindfoot to tibia peak angle (degree)	16.62 (2.34)	17.31 (3.09)	15.01 (1.31) *	22.79 (7.91)	14.92 (1.31)	9.32 (1.72) *	2.38 (−4.21 to 8.98)	5.68 (1.43 to 9.94) *	0.011	0.017	0.038^*^
Forefoot to hindfoot ROM (degree)	17.82 (2.75)	16.08 (1.86)	14.62 (0.79)	14.23 (0.99)	14.40 (1.79)	13.43 (0.69)	1.68 (−3.38 to 6.75)	1.18 (−0.88 to 3.25)	0.191	0.283	0.658
Forefoot to hindfoot peak angle (degree)	8.62 (1.35)	8.15 (0.80)	8.75 (1.20)	14.24 (6.17)	7.84 (1.27)	7.48 (1.13)	0.30 (−2.64 to 3.26)	1.26 (−1.97 to 4.51)	0.595	0.411	0.338
Hallux to forefoot ROM (degree)	23.34 (2.23)	26.03 (2.49)	28.76 (1.70) ^#^	21.06 (1.93)	21.07 (1.10)	21.65 (1.62) ^#^	4.96 (−0.37 to 10.30)	7.10 (2.48 to 11.73) ^#^	0.028 ^#^	0.186	0.346
Hallux to forefoot peak angle (degree)	22.06 (1.67)	26.78 (1.49) ^a^	27.12 (2.19) ^a^	21.94 (2.11)	21.15 (1.79) ^a^	20.50 (1.85) ^a^	5.63 (1.05 to 10.21) ^a^	6.61 (0.98 to 12.24) ^a^	0.049 ^a^	0.402	0.073
Maximum arch height (cm)	11.05 (0.35)	10.58 (0.26) ^b^	11.49 (0.89)	11.70 (0.38)	12.19 (0.60) ^b^	11.24 (0.44)	−1.61 (−2.91 to −0.31) ^b^	0.25 (−1.70 to 2.21)	0.049 ^b^	0.198	0.139
Minimum arch height (cm)	8.78 (0.34)	8.38 (0.37) ^c^	8.60 (0.79)	9.75 (0.34)	9.87 (0.51) ^c^	8.95 (0.41)	−1.48 (−2.73 to −0.23) ^c^	−0.34 (−2.10 to 1.41)	0.044 ^c^	0.476	0.327

^&^ interaction effect *p* = 0.049, between-group difference at 12 weeks (post hoc *p* = 0.013) and 24 weeks (post hoc *p* = 0.014). * interaction effect *p* = 0.038, between-group difference at 24 weeks (post hoc *p* = 0.009). ^#^ group effect *p* = 0.028, between-group difference at 24 weeks (post hoc *p* = 0.003). ^a^ group effect *p* = 0.049, between-group difference at 12 weeks (post hoc *p* = 0.016) and 24 weeks (post hoc *p* = 0.021). ^b^ group effect *p* = 0.049, between-group difference at 12 weeks (post hoc *p* = 0.015). ^c^ group effect *p* = 0.044, between-group difference at 12 weeks (post hoc *p* = 0.020).

## Data Availability

Data are owned by the Laboratório de Biomecânica do Movimento e Postura Humana-LaBiMPH, Departamento de Fisioterapia, Fonoaudiologia e Terapia Ocupacional, Faculdade de Medicina da Universidade de São Paulo. The data presented in this study are available on request from the corresponding author (icnsacco@usp.br). It is worth mentioning that this proof of concept analyzed preliminary data from a main RCT, and after its conclusion, the data will be available for access in the data repository of the University of São Paulo, which can be accessed at the following link: https://repositorio.uspdigital.usp.br/?codmnu=9980, accessed on 5 December 2022.

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
