# Peer review of "Could an Internet-Based Foot–Ankle Therapeutic Exercise Program Modify Clinical Outcomes and Gait Biomechanics in People with Diabetic Neuropathy? A Clinical Proof-of-Concept Study"

_sensors, 2022, doi:10.3390/s22249582_

Round 1

Reviewer 1 Report

The paper presents novel findings on the use of internet-based foot-ankle therapeutic exercises for people with diabetic neuropathy, which is part of a wider study. The findings of this study would be of interest to readers of the journal.

Abstract

“changed forefoot role during gait” this sentence requires clarification I am unclear if “role” is referring to rollover or load?

Introduction

Suggest revising “motion quality” to “quality of motion”

Methods

Clarification is required on the meaning of “educational orientations”. I suggest including a copy of the flyer used in the study as supplementary material.

“The main physiotherapist supervised all of the other 35 sessions remotely via the SOPeD interface” – please explain how the physiotherapist “supervised” the participants remotely?

Is it unclear why DPN severity was chosen as a primary outcome measure, while I understand interventions can influence DPN symptoms I don’t consider it is possible to reverse the severity of DPN?

Results

“Participant flow, attendance at follow-up assessment visits, and reasons for dropout are presented in Figure 2.” – I think this should be Figure 1 not 2?

The intervention group appear to have a longer time of onset of diabetes compared to the control group (18.8 vs. 10.8 years), do you consider that this may have had an impact on the results?

I suggest you comment on how the adherence rate of 62.0% compares to previous studies examining the use of internet-based interventions. Did the participants provide any feedback on why they did not complete all exercise sessions?

“attention must be given to keeping plantar loadings under the proper pressure threshold in foot areas at higher risk for ulceration” – the meaning of “under the proper pressure threshold” is unclear.

Author Response

RESPONSES TO THE REVIEWERS' COMMENTS ON A POINT-BY-POINT BASIS

Reviewer 01

We thank the reviewer for your comments and commitment in reviewing this manuscript. Below, we respond to your questions. If we have not yet been clear enough in our responses and arguments in this letter, as well as throughout the manuscript, we would like to have a new opportunity to better addressing your further questions or comments. First, we bring back your comments (in bold) and, subsequently, we provide our response. In the paper, we underlined the changes for you to follow what we improved in this new version.

The paper presents novel findings on the use of internet-based foot-ankle therapeutic exercises for people with diabetic neuropathy, which is part of a wider study. The findings of this study would be of interest to readers of the journal.

Response: Thank you very much for acknowledging that.

1) Abstract: “changed forefoot role during gait” this sentence requires clarification I am unclear if “role” is referring  to rollover or load?

Response:  In order to make it clear, the sentence has been rewritten.

“The IG showed improvements in the ankle and first metatarsophalangeal joint motion after 12 and 24 weeks, changed forefoot load absorption during foot rollover during gait after 24 weeks, reduced foot pain after 12 weeks, and improved foot function after 24 weeks.”

2) Introduction: Suggest revising “motion quality” to “quality of motion”.

Response: The writing has been revised. Thank you for the suggestion.

‘’As a result of these disorders, musculoskeletal and biomechanical alterations arise in daily living activities, affecting the quality of motion and functional performance.’’

3) Methods: Clarification is required on the meaning of “educational orientations ”. I suggest including a copy of the flyer used in the study as supplementary material.

Response: Thank you for pointing it out. We included the flyer used in the study as supplementary material.

“These self-care guidelines, adjusted for our study, were printed on a flyer given to all participants and included educational orientations (Supplementary Material – Flyer of self-care based on IWGDF).”

‘Supplementary Material: Flyer of self-care based on IWGDF given to all participants and included educational orientations.’’

4) Methods: “The main physiotherapist supervised all of the other 35 sessions remotely via the SOPeD interface” – please explain how the physiotherapist “supervised” the participants remotely?

Response: Thank you for your concern and for pointing it out. We included a sentence to make it clear. Please check below the addition made:

“Participants in the intervention group received access to use SOPeD that aims to provide self-care and allows the user to choose the best and most convenient time to carry out the exercise sessions. The exercise sessions were not monitored synchronously, but the main researcher could have access to SOPED administrator, at any time, to monitor how many accesses and how many exercise sessions were performed by each participant.’’

5) Methods: Is it unclear why DPN  severity was chosen as a primary outcome measure, while I understand interventions can influence DPN symptoms I don’t consider it is possible to reverse the severity of DPN?

Response: Thank you for your concern and for pointing it out. We evaluated DPN severity using the Decision Support System for Classification of Diabetic Polyneuropathy that includes three different input to compose a more comprehensive score about the neuropathy status: (1) symptoms extracted from the Brazilian version of the Michigan Neuropathy Screening Instrument (MNSI-BR), (2) vibration sensitivity assessed by a tuning fork (128 Hz), and (3) tactile sensitivity measured by a 10-g monofilament in 8 plantar areas. We chose the DPN severity as a primary outcome because it is a more comprehensive score including symptoms and signs of the DPN, compared to only use DPN symptoms. In addition, several studies (referenced below) have already shown that both signs and symptoms were modified when foot-related exercises were used as a therapeutic strategy, and thus its inclusion is worthwhile.

  • AHMAD, Irshad et al. Sensorimotor and gait training improves proprioception, nerve function, and muscular activation in patients with diabetic peripheral neuropathy: A randomized control trial. Journal of Musculoskeletal & Neuronal Interactions, v. 20, n. 2, p. 234, 2020.
  • KANCHANASAMUT, Wararom; PENSRI, Praneet. Effects of weight-bearing exercise on a mini-trampoline on foot mobility, plantar pressure and sensation of diabetic neuropathic feet; a preliminary study. Diabetic foot & ankle, v. 8, n. 1, p. 1287239, 2017.
  • MONTEIRO, Renan L. et al. Foot–ankle therapeutic exercise program can improve gait speed in people with diabetic neuropathy: a randomized controlled trial. Scientific reports, v. 12, n. 1, p. 1-12, 2022.
  • KLUDING, Patricia M. et al. Physical training and activity in people with diabetic peripheral neuropathy: paradigm shift. Physical therapy, v. 97, n. 1, p. 31-43, 2017.

We also included a sentence in the methods to make our choice clear:

‘’The primary outcomes were DPN symptoms and severity, as both were subject to improvements after therapeutic foot-ankle exercises [21,37], are important modifiable factors and DPN severity is an outcome assessed using a more comprehensive score than just using symptoms as outcomes, which includes DPN symptoms and signs (vibration and tactile sensitivities).’’

6) Results: “Participant flow, attendance at follow-up assessment visits, and reasons for dropout are presented in Figure 2.” – I think this should be Figure 1 not 2 ?

Response: Thank you for pointing it out. The correct is Figure 1.

We corrected it in the sentence:

“Participant flow, attendance at follow-up assessment visits, and reasons for dropout are presented in Figure 1.”

7) Results: The intervention group appear to have a longer  time of onset of diabetes compared to the control group (18.8 vs. 10.8 years), do you consider that this may have had an impact on the results?

Response: Thank you for your concern and for pointing it out. We agree that there may indeed be an influence of the difference in the time of onset of diabetes between the participants in the control and intervention groups, but as this is a preliminary study that does not include the complete study sample, we believe it is not advise to perform a covariance analysis with this sample (we use the Generalized Estimating Equation). But your point is very welcome, interesting, and in the final RCT study we will certainly perform a covariance analysis.

8) Results: I suggest you comment on how the adherence rate of 62.0% compares to previous studies examining the use of internet-based interventions. Did the participants  provide any feedback on why they did not complete all exercise sessions?

Response: Thank you for your concern and for pointing it out. We included a sentence to address your suggestion:

‘’An analysis of 101 papers conducted as part of a systematic review with the goal of analyzing adherence to web-based interventions revealed an average adherence rate of 50%, confirming that non-adherence is a problem with web-based interventions. The level of adherence varied greatly, with six programs scoring less than 10% and only five interventions achieving 90% or more [58].’’

‘’In this study, the reported reasons for not following the online program were internet problems and a broken cell phone.’’

9) “attention must be given to keeping plantar loadings under the proper pressure threshold  in foot areas at higher risk for ulceration” – the meaning of “under the proper pressure threshold” is unclear.

Response: Thank you for your concern and for pointing it out. We modified the sentence to make it clear. Please check below the change made:

‘’Although the increased pressure-time integral at the medial forefoot in the IG participants might represent a functional improvement in gait, attention must be given to keeping plantar loadings under a safe pressure range in foot areas at higher risk for ulceration [54].’’

Reviewer 2 Report

The paper is about a 12-week internet-based foot-ankle exercise program using  the SOPeD software. It has good impact on patients with diversified benefits.

The work has uniqueness.

But this paper needs extensive english grammar check.

Author Response

RESPONSES TO THE REVIEWERS' COMMENTS ON A POINT-BY-POINT BASIS

Reviewer 02

We thank the reviewer for the comment about the language in the manuscript. We would like to highlight that the manuscript has been revised by a native English reviewer from a Canadian professional Service (Scribendi), as we always do with our papers. We resubmitted the paper to English revision with the same service to check if this version is within their standard, and they sent us the Certificate of Editing and Proofreading, which we uploaded to the Sensors Journal System so that editors can have access. The certificate is attached in the "Author's note file". 

Reviewer 3 Report

Dear authors, first let me congratulate for the kind research and results displayed in the current paper. The results of the study will help clinicians to implement tele health and improve diabetes patients care.

I just note some minor comments before recommend it for publication:

-Abstract:

Please add some minor characteristics of included patients, I recommend to add IWGDF risk stratification classification.

- Introduction and rationale: the authors clearly described what is previously published about the topic under research. I propose to change the way the authors described the main outcome as follow: The principal aim of the study was to evaluate the clinical efficacy of (intervention) in clinical and biomechanical changes in persons with diabetes and DPN. It will empower your research, despite being a pilot RCT.

- Research design and methods: authors describe allocation and intervention properly.

I have just one question, how sample size calculation was reached? Authors declare that the results of the current pilot RCT are based in 30 DM participantes, despite this, which was the primary sample size calculation of the RCT? Based on a pilot study you do not need to add this to methods section, but it will improve methods section.

- Did patients wore therapeutic footwear and custom made insole to decrease PPP and PTI during the study? Please if possible add to research and methods section. 

- Did the authors evaluated adherence to perform exercises in IG? How was it evaluated? Explain please.

Results: 

- The results from the current research confirm previous data from other research groups, the power of the results remain in the telehealth reducing costs and visits for patients. Foot pain and PTI in the foot improved in the IG. 

Discussion:

- Authors should go further and link the current results with recurrence in high risk patients (IWGDF risk-3). 

- Please add external validity to the discussion section, what can influence the results in the clinical practice?

Author Response

RESPONSES TO THE REVIEWERS' COMMENTS ON A POINT-BY-POINT BASIS

Reviewer 03

We thank the reviewer for your comments and commitment in reviewing this manuscript. Below, we respond to your questions. If we have not yet been clear enough in our responses and arguments in this letter, as well as throughout the manuscript, we would like to have a new opportunity to better addressing your further questions or comments. First, we bring back your comments (in bold) and, subsequently, we provide our response. In the paper, we underlined the changes for you to follow what we improved in this new version.

Abstract: Please add some minor characteristics of included patients, I recommend to add IWGDF risk  stratification classification.

Response: Thank you for pointing it out. We included this information in the abstract:

 “We randomized 30 individuals with DPN (IWGDF risk category 1 or 2).”

Introduction: the authors clearly described what is previously published about the topic under research. I propose to change the way the authors described the main outcome as follow: The principal aim of the study was to evaluate the clinical efficacy of (intervention) in clinical and biomechanical changes in persons with diabetes and DPN. It will empower your research, despite being a pilot RCT.

Response: Thank you for pointing it out. As suggested, we rewrote the sentence of the study objective trying to empower our research, but as it is a proof of concept, we kept the information that it is a study that generates preliminary evidence on efficacy.

‘’The proposed clinical proof-of-concept study pursues preliminary evidence for potential efficacy of a 12-week internet-based foot-ankle therapeutic exercise program for people with diabetes and DPN to promote clinical and gait biomechanical changes.’’

Methods: authors describe allocation and intervention properly. I have just one question, how sample size calculation was reached? Authors declare that the results of the current pilot RCT are based   in 30 DM participantes, despite this, which was the primary sample size calculation of the RCT? Based on a pilot study you do not need to add this to methods section, but it will improve methods section.

Response: Thank you for your concern and for pointing it out. We included a sentence to make it clear. Please check below the addition made:

 “The main trial sample size was calculated using two important outcomes for patients with DPN. Considering the primary outcome (DPN symptoms), a medium effect size (0.52) was adopted and, for the secondary outcome (peak pressure at forefoot), a small effect size (0.20) was adopted. In order to obtain the largest sample size, the smallest effect size (0.20) was used. A statistical design of F-test repeated measures and interaction be-tween and within factors with two repeated measures and two study groups, a statistical power of 0.80, an alpha of 0.05, and an effect size of 0.20 were used for the sample size calculation. The resulting sample size was 52 individuals. A final sample size of 62 patients was then chosen after estimating a drop-out rate of 20%.”

Methods: Did patients wore therapeutic footwear and custom made insole to decrease PPP and PTI during the study? Please if possible add to research and methods section.

Response: Thank you for your concern and for pointing it out. We included a sentence to address your suggestion:

‘‘According to IWGDF recommendations, the use of therapeutic footwear or a custom-made insole is mandatory for patients with high ulcer risk (IWGDF category 3). As there is no mandatory prescription for low and moderate ulcer risk patients (IWGDF categories 1 or 2), the use of therapeutic footwear or a custom-made insole was not prescribed for the participants included in this study.’’

Methods: Did the authors evaluated adherence to perform exercises in IG? How was it evaluated? Explain please.

Response: Yes, adherence to perform exercises in IG was evaluated and this was already described in the methods section (item 2.3), as you can see below:

‘’The average number of sessions completed (36 sessions total) by IG participants was used to calculate the adherence to the program [36]. The number of all completed sessions was obtained from the SOPeD user databank and computed, even if the participant did not complete the full set of exercises in a given session’’.

Results: The results from the current research confirm previous data from other research groups, the power of the results remain in the telehealth reducing costs and visits for patients. Foot pain and PTI in the foot improved in the IG.

Response: Thank you very much for acknowledging and highlighting that.

Discussion: Authors should go further and link the current results with recurrence in high risk patients  (IWGDF risk-3).

Response: Thank you for pointing it out. We included a sentence to address your suggestion:

‘’The study was carried out with participants with ulcer risk (IWGDF categories 1 or 2) and presented positive results that we believe may even contribute to the prevention of the development of foot ulcers and amputations. However, it should be noted that further studies are needed with patients with ulcer risk (IWGDF category 3), since we did not test the intervention in this population.’’

Discussion: Please add external validity to the discussion section, what can influence the results in the clinical practice?

Response: Thank you for pointing it out. We included a sentence to address your suggestion:

‘’These results suggest that this intervention has benefits for people with DPN (low and moderate ulcer risk), and therefore its use in clinical practice is promising.’’

Round 2

Reviewer 2 Report

Now review comments are addressed.

Author Response

We thank you for the statement "Now review comments are addressed." We are glad that our manuscript now sounds checked regarding language.